# Agreement between the laboratory-based and non-laboratory-based WHO cardiovascular risk charts: a cross-sectional analysis of a national health survey in Peru

Wilmer Cristobal Guzman-Vilca [1,2] Gustavo A Quispe-Villegas,[1] Fritz Fidel Váscones Román,[1] Antonio Bernabe-Ortiz [2,3] Rodrigo M Carrillo-Larco [2,4,5]

For numbered affiliations see end of article.

**Correspondence to**
Dr Rodrigo M Carrillo-Larco; rcarrill@ic.ac.uk

## ABSTRACT

**Objective** To determine the agreement between the cardiovascular disease (CVD) risk predictions computed with the WHO non-laboratory-based model and laboratory-based model in a nationally representative sample of Peruvian adults.

**Design** Cross-sectional analysis of a national health survey.

**Methods** Absolute CVD risk was computed with the 2019 WHO laboratory-based and non-laboratory-based models. The risk predictions from both models were compared with Bland-Altman plots, Lin's concordance coefficient correlation (LCCC), and kappa statistics, stratified by sex, age, body mass index categories, smoking and diabetes status.

**Results** 663 people aged 30–59 years were included in the analysis. Overall, there were no substantial differences between the mean CVD risk computed with the laboratory-based model 2.0% (95% CI 1.8% to 2.2%) and the non-laboratory-based model 2.0% (95% CI 1.8% to 2.1%). In the Bland-Altman plots, the limits of agreement were the widest among people with diabetes (−0.21; 4.37) compared with people without diabetes (−1.17; 0.95). The lowest agreement as per the LCCC was also seen in people with diabetes (0.74 (95% CI 0.63 to 0.82)), the same was observed with the kappa statistic (kappa=0.36). In general, agreement between the scores was appropriate in terms of clinical significance.

**Conclusions** The absolute cardiovascular predicted risk was similar between the laboratory-based and non-laboratory-based 2019 WHO cardiovascular risk models. Pending validation from longitudinal studies, the non-laboratory-based model (instead of the laboratory-based) could be used when assessing CVD risk in Peruvian population.

## STRENGTHS AND LIMITATIONS OF THIS STUDY

⇒ This analysis provided the first evidence of the agreement between the 2019 WHO CVD risk laboratory-based and non-laboratory-based models.

⇒ We leveraged on the most recent nationally representative survey that included blood biomarkers in Peru.

⇒ Our study population was young, mostly women, and with overall low absolute cardiovascular predicted risk. No one in the study population had an absolute cardiovascular risk ≥20%.

⇒ We assumed all participants were free of cardiovascular disease (CVD) to use the 2019 WHO CVD risk score, as information regarding history of cardiovascular events was not reported.

CVDs impose a huge burden in low and middle-income countries (LMICs),[4] where deaths from CVDs occur at younger ages compared with high-income countries (HICs).[1] However, CVD can be prevented and managed through a combination of population-level and individual-level interventions;[5] for the latter, the identification of individuals at high cardiovascular risk is a cornerstone in the prevention of CVD. In this line, the WHO and the Pan American Health Organization (PAHO),[6] alongside several clinical guidelines,[7–9] recommend CVD risk stratification with CVD risk prediction models to inform evidence-based treatment.

CVD risk prediction models identify people who would benefit the most of preventive interventions (eg, statin therapy).[10] Although there are several CVD risk prediction models,[11] these were mostly developed in HICs limiting their application in LMICs where they would need recalibration to deliver accurate predictions to guide treatment allocation. To

## INTRODUCTION

Cardiovascular diseases (CVDs) are the main cause of death globally.[1] In 2019, CVD caused 18.5 million deaths in adults, representing 36% of all global deaths.[2 3] Furthermore,

overcome this limitation, the WHO convened a global effort to derive, calibrate and validate new CVD risk prediction models for all world regions.[12] Two WHO CVD risk prediction models were developed: a laboratory-based model and a non-laboratory-based model. Because laboratory biomarkers (eg, total cholesterol) may not be available in primary healthcare centres in LMICs limiting the use of laboratory-based CVD risk prediction models,[13] the non-laboratory-based model arises as a handy tool for clinicians in LMICs. Similarly, CVD risk prediction models are used to monitor the prevalence of high CVD risk and treatment coverage (ie, people at high CVD receiving treatment).[14] In this context, countries conducting national or large population-based health surveys without lipid biomarkers could benefit from the non-laboratory-based models. Peru, for example, does not have regular national health surveys including total cholesterol, but Peru has a yearly national health survey including anthropometrics, blood pressure and health questionnaires. In Peru, and other similar LMICs, it would not be possible to monitor the burden of high CVD risk with a laboratory-based model and the non-laboratory-based model rises as the only alternative. Nonetheless, evidence regarding the agreement between the WHO laboratory-based and non-laboratory-based models in countries from Latin America is missing.[12] Whether the WHO non-laboratory-based model delivers predictions similar to those of the WHO laboratory-based model remains unknown in Latin America. However, clinicians need this evidence to inform their choice (non-laboratory-based vs laboratory-based model), and to interpret the results of the non-laboratory-based model under the assumption that the laboratory-based model is the gold standard.[12] To provide this evidence for practitioners in Peru, we determined the agreement between the risk predictions computed with the non-laboratory-based model and laboratory-based model in a nationally representative sample of Peruvian adults.

## METHODS

### Data sources

This is a cross-sectional study of a national survey conducted by the National Centre for Food and Nutrition (CENAN, for its acronym in Spanish) of Peru. CENAN's survey was conducted between 2017 and 2018 on a nationally representative sample of Peruvian adults aged between 18 and 59 years.[15] Of note, this is the most recent nationally representative survey conducted in Peru that included blood biomarkers (eg, lipid profile). CENAN's survey adhered to ethical guidelines and followed a standardised protocol that has been published elsewhere.[15] Each participant was informed about all procedures and techniques used in the survey; also, participants could have left the study at any time and their personal information was kept confidential.[15]

The CENAN's survey sample was computed using the formula shown in online supplemental figure 1 and followed a probabilistic sampling design approach with two stages.[15] First, clusters were randomly selected considering three strata: (1) urban areas except Lima city, (2) rural areas and (3) Lima city. Then, households (of adults aged 18–59 years living in) were randomly selected within each cluster. To be selected for the survey sample, participants had to fulfil the following inclusion criteria: (1) adults aged 18–59 years and (2) fasting 9–12 hours for blood biomarkers. The following participants were excluded: (1) pregnant and postpartum women, (2) adults taking medication that could alter glucose and lipid profiles, (3) adults with congenital diseases that could limit anthropometrics measurement (eg, Down syndrome).

### Study population

We analysed a complete-case sample regarding all the laboratory-based and office-based 2019 10-year WHO CVD risk score variables (see the Variables section). We studied men and women aged between 30 and 59 years. The younger age limit was decided because CVD risk models are not recommended in younger individuals; the older age limit was decided because of the survey design. A flowchart of data cleaning is shown in online supplemental figure 2. We did not apply other selection criteria.

Although the 2019 10-year WHO CVD risk models were developed for people aged 40–80 years,[12] for people aged <40 years we assumed they had 40 years for the absolute CVD risk computation. This is consistent with the Package of essential non-communicable (PEN) disease interventions for primary healthcare in low-resource settings,[16] and was also done in a recent global work.[17]

### Variables

We calculated the 10-year CVD risk at the individual level following the laboratory-based and non-laboratory-based 2019 10-year WHO CVD risk model.[12] We used the *whocvdrisk* command in STATA, which was developed by the authors of the 2019 10-year WHO CVD risk charts.[18] For the laboratory-based model, scores were calculated based on: age (years), current smoking status (yes/no), systolic blood pressure (SBP, mm Hg), history of self-reported diabetes diagnosis (yes/no), and total cholesterol (mmol/L).[12] For the non-laboratory-based model, we used age (years), current smoking status (yes/no), SBP (mm Hg) and body mass index (BMI, kg/m$^2$).[12]

The CENAN's survey collected anthropometrics and three BP measurements that were taken by trained field-workers following a standard protocol.[15] We computed BMI using measured weight (kg) divided by the square of height (metres); for descriptive purposes, we classified BMI in three levels: normal weight (BMI <25 kg/m$^2$), overweight (BMI ≥25–29.9 kg/m$^2$) and obesity (BMI ≥30 kg/m$^2$). BMI records outside the range 10–80 kg/m$^2$ were discarded. As the third BP measurement was only available in few participants (<2% of the initial sample), we used the second SBP measurement only (ie, the first and third SBP records were discarded in the main analysis).

Of note, there were no substantial differences between the first and second SBP records (online supplemental table 1); nonetheless, we performed a sensitivity analysis using the mean SBP of first and second SBP records. We discarded any SBP records outside the range 70–270 mm Hg.

For people who self-reported being under antihypertensive treatment, we used the pre-treatment SBP; this is consistent with the PEN protocol and with a previous global work.[16 17] Pre-treatment SBP was computed as: pre-treatment systolic blood pressure = (current systolic blood pressure−6.3)/0.9.[19] Conversely, for those not taking antihypertensive treatment, we used the recorded SBP as was.

We defined current smoker with one question coded as no versus yes: *Do you currently smoke any tobacco product such as cigarettes, cigars or pipes?* Self-reported information about a prior history of diabetes was assessed by a question also coded as no vs yes: *Have you ever been told by a physician or another healthcare worker that you have high blood sugar or diabetes?*

Total cholesterol was obtained via enzymatic colorimetric method.[15] Because CENAN's survey data had total cholesterol in mg/dL, these values were divided by 38.67 to obtain total cholesterol in mmol/L.

### Statistical analysis

We determined the agreement between the absolute CVD risk predicted with the WHO laboratory-based and non-laboratory-based models following three methods: Bland-Altman plots, Lin's concordance coefficient correlation (LCCC) and kappa statistic. We considered the absolute CVD risk as a continuous variable, and the agreement between both models was examined using Bland-Altman plots and the LCCC. Furthermore, we considered the CVD risk as a categorical variable, and divided into three groups: <5%, 5–9% and 10–19%; because there were no observations in the high-risk category (CVD risk ≥20%), agreement was not examined in this group. For these categories, we evaluated the agreement using the kappa statistic. For the Bland-Altman plots, LCCC and the kappa statistic, results were stratified by sex, 10-year age groups, BMI categories, smoking status, self-reported diabetes diagnosis and urban/rural location.

In the Bland-Altman plots, the risk difference between the laboratory-based and non-laboratory-based absolute cardiovascular predicted risk was plotted on the vertical axis, and the mean of both scores on the horizontal axis.[20] As the true risk of CVDs at the individual level is uncertain, the mean of both scores is the best available estimate.[20 21] The 95% of the limit of agreement was represented by the mean difference of both scores±two SD; this limit provides an interval in which 95% of the differences between both scores would be expected to lie.[20] The LCCC between the laboratory-based and non-laboratory-based absolute cardiovascular predicted risks was also evaluated. The agreement based on the LCCC ranges between −1 and 1, with 1 suggesting a perfect agreement. The categorical agreement was evaluated using the Kappa statistic. Kappa

<0 indicated less than chance agreement, and values 0.01–0.20, 0.21–0.40, 0.41–0.60, 0.61–0.80 and 0.81–0.99 represented slight, fair, moderate, substantial and almost perfect agreement, respectively.[22]

All analyses code were conducted with R (V.4.0.3) and STATA (V.17.0, College Station, Texas). Population characteristics along with their 95% CI were summarised accounting for the complex survey design of the CENAN's survey.[15]

### Patient and public involvement

No patient involved.

## RESULTS

Our pooled dataset included 663 participants (online supplemental figure 2). The mean age was 44.0 years (95% CI 43.2 to 44.7) and the proportion of men was 41.5%. The mean SBP was 108.9 mm Hg (95% CI 107.5 to 110.3 mm Hg) and the mean BMI was 28.8 kg/m² (95% CI 28.3 to 29.2 kg/m²). The proportion of people with overweight was 41.0% (95% CI 36.6% to 45.6%), whereas 35.9% (95% CI (31.6% to 40.5%) of the population were obese. The mean total cholesterol was 4.9 mmol/L (95% CI 4.8 to 5.0 mmol/L), 11.7% (95% CI 8.8% to 15.3%) of the population were smokers and 7.1% (95% CI 5.0% to 9.8%) had diabetes (table 1).

### Absolute cardiovascular risk according to the 2019 WHO cardiovascular risk models

Overall, there were no substantial differences between the mean absolute cardiovascular risk computed with the laboratory-based and non-laboratory-based models (table 1). The mean absolute cardiovascular risk was 2.0% (95% CI 1.8% to 2.2%) according to the laboratory-based model, and 2.0% (95% CI 1.8% to 2.1%) according to the non-laboratory-based model. In both models, the mean absolute cardiovascular risk was higher in men than women. The sensitivity analysis (using the mean of two SBP records) yielded the same findings in the overall sample: 2.0% (95% CI 1.8% to 2.2%) in the laboratory-based-model and 2.0 (95% CI 1.8% to 2.1%) in the non-laboratory-based model.

### Mean difference between risk predictions

Overall, the mean difference between the laboratory-based and the non-laboratory-based models was 0.03% (95% CI −0.03% to 0.10%). According to sex, the mean difference between models was −0.02% (95% CI −0.14% to 0.09%) in men, and 0.08% (95% CI 0.01% to 0.15%) in women. According to age, the mean difference between models was −0.07% (95% CI −0.12% to 0.04%) in people aged 30–39 years, 0.04% (95% CI −0.08% to 0.15%) in people aged 40–49 years and 0.17% (95% CI 0.02% to 0.32%) in people aged 50–59 years. Stratified by BMI categories, the mean difference was 0.10% (95% CI 0.02% to 0.18%) in people with normal weight, 0.08% (95% CI 0.00% to 0.17%) in people with overweight, and −0.07%

**Table 1** Weighted distribution of the predictors in the 2019 WHO CVD risk models, overall and by sex

|  | Total | Men | Women |
| --- | --- | --- | --- |
| Sample size | 663 | 280 | 383 |
| Age (mean and 95% CI, years) | 44 (43.2 to 44.7) | 44.2 (43 to 45.4) | 43.8 (42.9 to 44.7) |
| Proportion of people aged 30–39 years (95% CI, %) | 35.8 (31.6 to 40.3) | 37.2 (31 to 43.9) | 34.8 (29.4 to 40.7) |
| Proportion of people aged 40–49 years (95% CI, %) | 34 (29.8 to 38.4) | 29.5 (24 to 35.7) | 37.1 (31.5 to 43.1) |
| Proportion of people aged 50–59 years (95% CI, %) | 30.2 (26.3 to 34.5) | 33.3 (27.1 to 40) | 28 (23.5 to 33.1) |
| Systolic blood pressure (mean and 95% CI, mm Hg) | 108.9 (107.5 to 110.3) | 116.1 (113.8 to 118.4) | 103.8 (102.3 to 105.3) |
| Diastolic blood pressure (mean and 95% CI, mm Hg) | 71.9 (71 to 72.8) | 74.6 (73.1 to 76.1) | 70 (69 to 71) |
| Body mass index (mean and 95% CI, kg/m$^2$) | 28.8 (28.3 to 29.2) | 28.3 (27.6 to 29) | 29.1 (28.5 to 29.7) |
| Proportion of people with normal weight (95% CI, %) | 23.1 (19.5 to 27.2) | 24.5 (19.5 to 30.4) | 22.1 (17.2 to 27.9) |
| Proportion of people with overweight (95% CI, %) | 41 (36.6 to 45.6) | 45.3 (38.9 to 52) | 37.9 (31.9 to 44.3) |
| Proportion of people with obesity (95% CI, %) | 35.9 (31.6 to 40.5) | 30.1 (24.1 to 36.9) | 40 (34 to 46.2) |
| Total cholesterol (mean and 95% CI, mmol/L) | 4.9 (4.8 to 5) | 4.8 (4.6 to 5) | 4.9 (4.8 to 5.1) |
| Proportion of smokers (95% CI, %) | 11.7 (8.8 to 15.3) | 21.1 (15.8 to 27.6) | 5 (3 to 8.3) |
| Proportion of people with diabetes (95% CI, %) | 7.1 (5 to 9.8) | 7.8 (4.7 to 12.6) | 6.5 (4.3 to 9.9) |
| Laboratory-based CVD risk score (mean and 95% CI, %) | 2 (1.8 to 2.2) | 2.6 (2.3 to 2.9) | 1.6 (1.4 to 1.7) |
| Non-laboratory-based CVD risk score (mean and 95% CI, %) | 2 (1.8 to 2.1) | 2.7 (2.4 to 3) | 1.5 (1.4 to 1.6) |

CVD, cardiovascular disease.

(95% CI −0.21% to 0.07%) in people with obesity. The mean difference was 0.05% (95% CI −0.34% to 0.44%) in smokers and 0.03% (95% CI −0.02% to 0.09%) in non-smokers. According to self-reported diabetes status, mean difference was 2.08% (95% CI 1.73% to 2.44%) in people with self-reported diabetes, and −0.11% (95% CI −0.15% to 0.07%) in people without self-reported diabetes. The sensitivity analysis provided similar results across all variables; for example, the largest mean difference was also observed in people with self-reported diabetes (2.09% (95% CI 1.73 to 2.45%).

### Bland-Altman plots and limits of agreement
The limit of agreement was slightly narrower for women (−1.23; 1.39) compared with men (−1.93; 1.89) (figure 1).

The limit of agreement widened with older ages and higher BMI levels; for example, the limit of agreement was narrower in people aged 30–39 years (−0.72; 0.56) compared with those aged 50–59 years (−1.96; 2.29) (figure 2), and for people with normal weight (−0.90; 1.09) compared with those with obesity (−2.15; 2.02) (figure 3). According to smoking and self-reported diabetes status, the limit of agreement was wider in those who had the condition. For example, the limit of agreement in smokers (−3.13; 3.24) was wider compared with non-smokers (−1.25; 1.32) (figure 4). Similarly, in people with self-reported diabetes (−0.21; 4.37), the limit of agreement was wider compared with people without self-reported diabetes (−1.17; 0.95) (figure 5). Notably, the

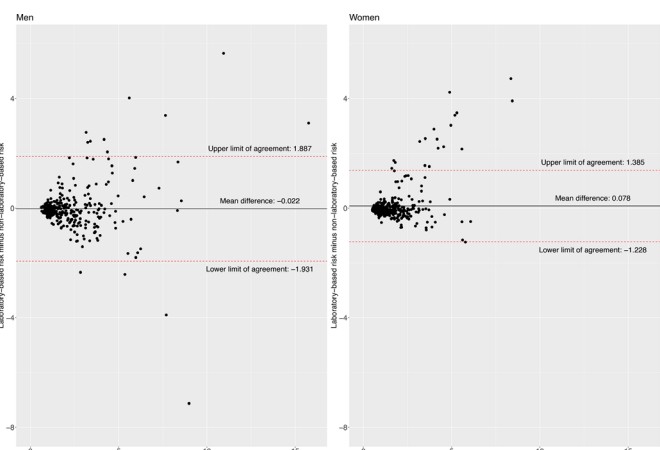

**Figure 1** Bland-Altman plots showing agreement between laboratory-based and non-laboratory-based risk scores according to sex.

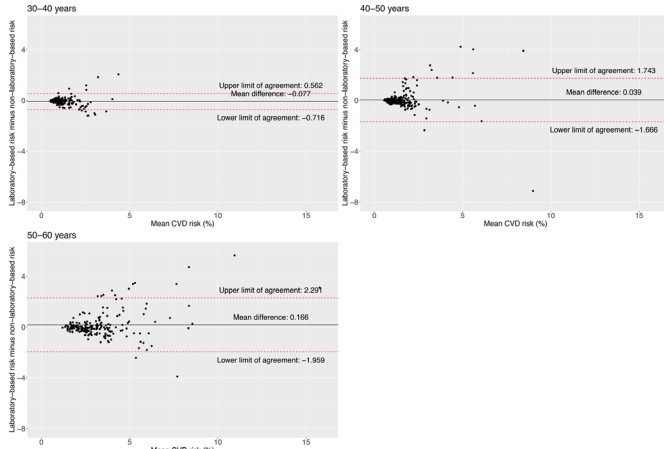

**Figure 2** Bland-Altman plots showing agreement between laboratory-based and non-laboratory-based risk scores according to age groups.

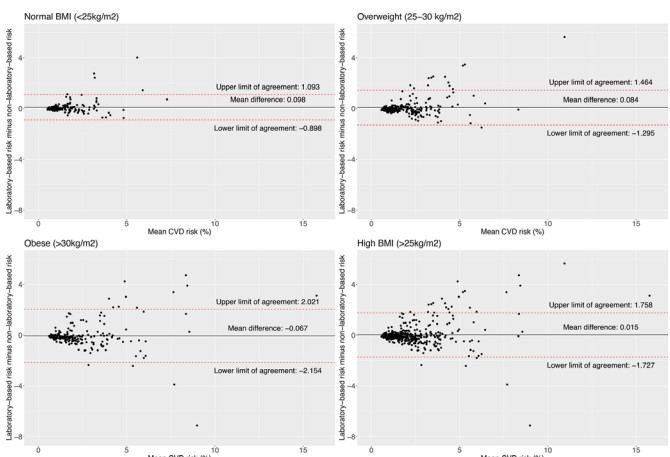

**Figure 3** Bland-Altman plots showing agreement between laboratory-based and non-laboratory-based risk scores according to body mass index (BMI) categories.

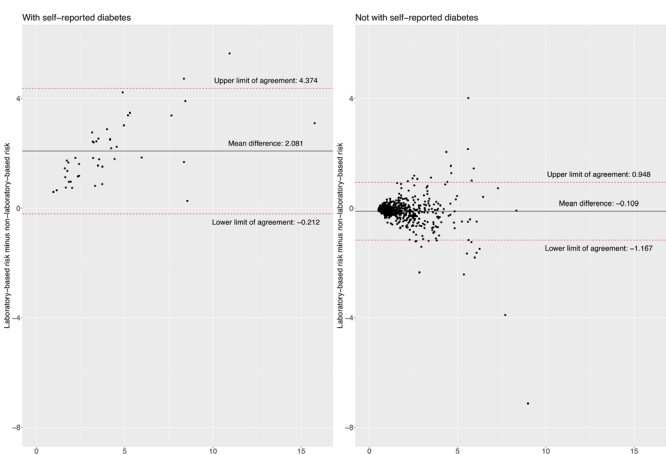

**Figure 5** Bland-Altman plots showing agreement between laboratory-based and non-laboratory-based risk scores according to self-reported diabetes status.

sensitivity analysis resulted in similar limits of agreement in all variables (online supplemental figures 3–7).

### Agreement by LCCC

The overall agreement between scores as per the LCCC was 0.87 (95% CI 0.85 to 0.89), and it was virtually the same in men (0.87 (95% CI 0.84 to 0.89)) and women (0.85 (95% CI 0.82 to 0.87)). Across age groups, the highest agreement was seen in the 30–39 age group (0.87 (95% CI 0.84 to 0.9)). Across BMI categories, it was the normal BMI category, which had the highest agreement (0.90 (95% CI 0.87 to 0.92)). Overall, the lowest agreement values were seen across smokers (0.81 (95% CI 0.72 to 0.88)), those aged 40–49 years (0.74 (95% CI 0.67 to 0.79)), and those with self-reported diabetes (0.74 (95% CI 0.63 to 0.82)) (table 2). Similar results were seen in the sensitivity analysis (online supplemental table 2).

### Categorical agreement

In the overall population, there was a slightly larger number of people categorised as having an absolute CVD

risk of 5–9% and 10–19% with the laboratory-based model compared with the non-laboratory-based model (online supplemental table 3). For example, the laboratory-based model categorised 37 people in the 5–9% CVD risk category, whereas the non-laboratory-based model categorised 26 people. Overall, the agreement between risk categories was substantial (kappa=0.62), and it was better for men (kappa=0.70) compared with women (kappa=0.44) (online supplemental table 4). Of note, the

**Table 2** Lin's concordance coefficient correlation showing agreement between laboratory-based and non-laboratory-based risk models according to the predictors in the 2019 WHO CVD risk models and urban/rural location

| Variables | Categories | Lin's concordance coefficient correlation (95% CI) |
|---|---|---|
| Sex | Men | 0.87 (0.84 to 0.89) |
| | Women | 0.85 (0.82 to 0.87) |
| Age (years) | 30–39 | 0.87 (0.84 to 0.9) |
| | 40–49 | 0.74 (0.67 to 0.79) |
| | 50–59 | 0.83 (0.78 to 0.86) |
| Body mass index category | Normal | 0.9 (0.87 to 0.92) |
| | Overweight | 0.87 (0.85 to 0.9) |
| | Obese | 0.86 (0.82 to 0.89) |
| Smoking status | Smoker | 0.81 (0.72 to 0.88) |
| | Non-smoker | 0.86 (0.84 to 0.88) |
| Diabetes status | With self-reported diabetes | 0.74 (0.63 to 0.82) |
| | Not with self-reported diabetes | 0.91 (0.9 to 0.92) |
| Urban or rural | Urban | 0.88 (0.85 to 0.9) |
| | Rural | 0.85 (0.82 to 0.88) |

CVD, cardiovascular disease.

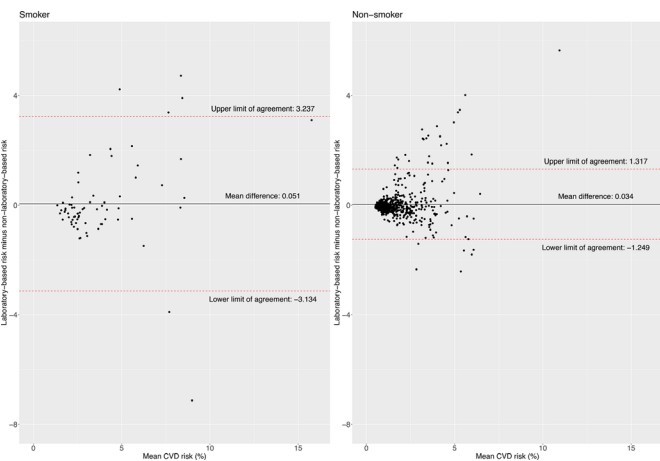

**Figure 4** Bland-Altman plots showing agreement between laboratory-based and non-laboratory-based risk scores according to smoking status. CVD, cardiovascular disease.

lowest agreement between risk categories was observed among people with self-reported diabetes (kappa=0.36): out of 14 people with self-reported diabetes in the 5–9% CVD risk category following the laboratory-based-model, 4 were placed in the same category following the non-laboratory-based model, as the rest were placed in the 0–5% CVD risk category. The categorical agreement according to all variables is presented in online supplemental tables 3–9.

## DISCUSSION
### Main findings
In this work, we evaluated the agreement between the CVD risk estimates predicted with the 2019 WHO 10-year laboratory-based and non-laboratory-based models in a nationally representative sample of Peruvian adults. The mean absolute predicted CVD risk according to both models in the general population was virtually the same. In addition, we found that the limits of agreement between both models increased with a higher CVD risk; for instance, the limits of agreement were wider in smokers and people with self-reported diabetes. We observed good agreement between the laboratory-based and non-laboratory-based models in terms of clinical significance.[21 23] These findings suggest that, in a population with a similar profile to that in this study, practitioners could use either the laboratory-based or non-laboratory-based models. Although the difference is very small, careful interpretation could be needed for people with cardiovascular risk factors: obesity, smokers and people with self-reported diabetes, among whom the difference was slightly larger than in their peers without these risk factors.

### Public health implications
We provided insights about the applicability of the 2019 WHO non-laboratory-based model in the Peruvian population. This evidence is relevant in terms of clinical practice and public health in Peru and other similar countries (eg, Andean Latin America) because it informs whether the predictions based on the laboratory-based and non-laboratory-based CVD risk models are equivalent. If so, either of these models could be used without substantial bias, hence supporting the use of the non-laboratory-based model when blood biomarkers are not available. The latter is of special importance in LMICs like Peru, where laboratory measurements are not always available in primary health centres, which represent>98% of all healthcare facilities in Peru.[24]

According to the three statistical analyses we implemented (Bland-Altman plots, LCCC; and kappa statistics), our results suggest that the agreement between the laboratory-based and non-laboratory-based models was appropriate among Peruvians with low CVD risk and younger than 60 years. In other words, our results suggest that the laboratory-based and non-laboratory-based models provide similar predictions and may therefore

be used interchangeably as needed, though the profile of our study population ought to be considered when extracting or implementing our findings into clinical practice and public health. Of note, the small number of participants in some variables of interest (eg, only 44 participants had self-reported diabetes) could have explained the broader limits of agreement in our results. Future studies should include a larger number of participants to further confirm whether the limits of agreement are wider according to smoking and diabetes status.

### Research in context
The study most comparable to ours evaluated the agreement between the Framingham 10-year CVD risk laboratory and non-laboratory models on a population aged 40–75 years in southern Iran.[23] They found the mean CVD risk following the non-laboratory-based model (9.4%) was higher than the laboratory-based model (6.7%).[23] Additionally, their limits of agreement between both Framingham models in people <60 years old were wider compared with ours in both men (−1.9–1.9 by our estimates vs −2.5%–8.9% by Rezaei et al[23]) and women (−1.2–1.4 by our estimates vs −2.3%–4.6% by Rezaei et al[23]). This could be explained by the fact that Rezaei et al[23] included an older population, which tend to have higher levels of CVD risk factors and therefore a higher absolute CVD risk. As limits of agreement between two models tend to widen with higher CVD risk,[21 23] our limits of agreement would presumably be wider if we had studied a similar population to that of the work by Rezaei et al.[23] The differences between our results could be further explained by the CVD risk score herein used. We used the 2019 WHO CVD risk models,[12] whereas Rezaei et al used the Framingham risk scores. The Framingham risk score was developed for a more specific population (Caucasians in the US),[25] yet the 2019 WHO CVD risk model was developed and recalibrated for a global use (eg, those living in LMICs).[12]

The agreement between the 2019 WHO laboratory-based and non-laboratory-based model was also explored in the global work convened by the WHO.[12] They applied the two models to WHO STEPS surveys and compared the proportion of people categorised at different levels of predicted CVD risk.[12] Overall, they found moderate agreement between both models, and their discrepancy was attributed to poor performance of the non-laboratory-based model in people with diabetes.[12] This finding is consistent with our results because we found the widest limits of agreement, the lowest LCCC, and the lowest categorical agreement in people with self-reported diabetes. When possible, it would seem reasonable to use the laboratory-based model in those whose have diabetes.

### Potential explanations
In our study, diabetes status was only included in the laboratory-based model (and not in the non-laboratory-based model) and reasonably, we observed the lowest agreement between both models in those with diabetes. That is, in people with diabetes, the non-laboratory-based

model underestimated the absolute CVD risk computed following the laboratory-based model. Probably because people with diabetes are already at high CVD risk and the non-laboratory model, we used without diabetes information would underestimate the absolute risk.

## Strengths and limitations

To the best of our knowledge, we provided the first evidence of the agreement between the 2019 WHO CVD risk laboratory-based and non-laboratory-based models; furthermore, we leveraged on a nationally representative survey conducted in a LMIC.[26] Different CVD risk equations have been created but none has proved to produce reliable estimates for LMICs. The 2019 WHO CVD risk charts were adapted for LMICs using extensive datasets for its derivation, recalibration and validation, which brings them several advantages over previous risk charts. Nonetheless, this study has also limitations. First, our study population was young (30–59 years), mostly women (58%), and with overall low absolute cardiovascular predicted risk. This led to the observation that no one in the study population had an absolute cardiovascular risk ≥20%. Thus, we could only draw conclusions for people within the low and medium CVD risk range, and with a similar demographic and risk factor profile. We acknowledge that further subgroup analysis could be relevant, for example by diabetes status. However, because of data availability and the reduced number of observations in some groups, this subgroup analysis would be impossible to conduct. Future work in Peru and Latin America should verify our results with a larger, older and more diverse population. Second, as CENAN's survey did not include history of cardiovascular events, we assumed all participants were free of CVD to use the 2019 WHO CVD risk score. This approach could have led to higher absolute cardiovascular risk because people who have had a cardiovascular event (eg, myocardial infarction) are at higher risk of another cardiovascular event. Nonetheless, considering that our study population was young and therefore with a low incidence of cardiovascular diseases,[3] the proportion of people with history of CVD excluded from the total sample size would have been small; not excluding potentially a small group may not have altered the overall results. Third, we only used one blood pressure record (the second SBP out of two measurements). Ideally, and following standard protocols recommended by WHO and other international organisations,[27 28] we should have used the average of multiple records having discarded the very first measurement. This was not possible with the available data because they only measured blood pressure twice. Nonetheless, we performed a sensitivity analysis using the mean SBP of the first and second records and had virtually the same results as in our main analysis in which only the second SBP record was used.

## CONCLUSIONS

The absolute cardiovascular predicted risk was similar between the laboratory-based and non-laboratory-based 2019 WHO cardiovascular risk models. Pending validation from longitudinal studies, the non-laboratory-based model (instead of the laboratory-based which requires additional resources) could be used in Peruvian population. Nonetheless, it should be noted that the agreement between these models was less clear in people with cardiovascular risk factors: obesity, smokers and people with diabetes. While universal health coverage momentum helps to have laboratory tests in (most) primary care facilities to use the laboratory-based model, it seems reasonable to use the non-laboratory-based model for primary prevention of CVD following the risk stratification approach.

**Author affiliations**
[1]School of Medicine 'Alberto Hurtado', Universidad Peruana Cayetano Heredia, Lima, Peru
[2]CRONICAS Centre of Excellence in Chronic Diseases, Universidad Peruana Cayetano Heredia, Lima, Peru
[3]Universidad Cientifica del Sur, Lima, Peru
[4]Department of Epidemiology and Biostatistics, Imperial College London, London, UK
[5]Universidad Continental, Lima, Peru

**Contributors** RMC-L conceived the idea with WCG-V. WCG-V conducted the analysis with support from GAQ-V and FFVR. WCG-V wrote the first draft of the manuscript with support from GAQ-V, FFVR, AB-O and RMC-L. All authors provided relevant scientific contribution and approved the submitted version. RMC-L and WCG-V are the guarantors of this study, and accept full responsibility for the work, had access to the data and controlled the decision publish.

**Funding** RMC-L is supported by a Wellcome Trust International Training Fellowship (214185/Z/18/Z).

**Competing interests** None declared.

**Patient and public involvement** Patients and/or the public were not involved in the design, or conduct, or reporting, or dissemination plans of this research.

**Patient consent for publication** Not applicable.

**Ethics approval** Not applicable.

**Provenance and peer review** Not commissioned; externally peer reviewed.

**Data availability statement** Data may be obtained from a third party and are not publicly available. Datasets available upon request from the CENAN.

**ORCID iDs**
Wilmer Cristobal Guzman-Vilca http://orcid.org/0000-0002-2194-8496
Fritz Fidel Váscones Román http://orcid.org/0000-0001-9564-0710
Antonio Bernabe-Ortiz http://orcid.org/0000-0002-6834-1376
Rodrigo M Carrillo-Larco http://orcid.org/0000-0002-2090-1856

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
