## [Reviewer comments · BMJ Open]

ARTICLE DETAILS

TITLE (PROVISIONAL)	Agreement between the laboratory- and non-laboratory-based WHO cardiovascular risk charts: a cross-sectional analysis of a national health survey in Peru
AUTHORS	Guzman-Vilca, Wilmer Cristobal; Quispe-Villegas, Gustavo A.; Váscones Román, Fritz Fidel; Bernabe-Ortiz, Antonio; Carrillo-Larco, Rodrigo

VERSION 1 – REVIEW

REVIEWER	Boateng, Daniel UMC Utrecht
REVIEW RETURNED	30-Jun-2022

GENERAL COMMENTS	This study adds to the evidence of support CVD risk prediction for the prevention of CVD related morbidity and mortality. Kindly find below some comments: - Please add a recommendation to the conclusion of the abstract. What you have currently speaks more to the findings. What do you make out of these findings in terms of policy?- In the strength and limitations, do you consider the third point as a strength or limitation?- You stated that: “Our results suggest that the agreement between the laboratory-based and non-laboratory-based models was appropriate among Peruvians with low CVD risk and younger than 60 years”- How did you show the appropriateness of this comparisons in your study?- Authors failed to show how justify the importance of this study, and the need for comparing the two algorithms in this context. Many studies have compared CVD algorithms. Why is the comparison of these two important, especially for your population?- There was little explanation for findings of this study in the discussion.
--

REVIEWER	Mirzaei, Mohsen Shahid Sadoughi University of Medical Sciences and Health Services, Community Medicine
REVIEW RETURNED	01-Jul-2022

GENERAL COMMENTS	Thanks to the authors for comparing the two methods (there are laboratory limitations in many areas). Findings from such studies lend credence to non-laboratory assessments. The answers to these questions will help other researchers to better analyze or conduct a similar study:  - Briefly explain how to sample the population in the national study. -The inclusion and exclusion criteria of the study should be stated. Not considering a history of previous cardiovascular disease confounds the final analysis. This group is at high risk. Although the limitations are mentioned, it is necessary to discuss more. - How is the sample size calculated? -This study used national survey data. The approval of the ethics committee for the use of data should be noted. - In the discussion, it is necessary to compare the results of this study with similar studies. Has the agreement between the two methods been reported in several studies in other regions? Has a study been published that reported the difference in the results of the two methods? If yes, has the study method been different? Comparison with the findings of others is recommended.
---

VERSION 1 – AUTHOR RESPONSE

REVIEWER #1

We appreciate **Dr Boateng** took the time to comment on our work. We have incorporated his suggestions.

Q1. Please add a recommendation to the conclusion of the abstract. What you have currently speaks more to the findings. What do you make out of these findings in terms of policy?

A1. We have modified our conclusion. This now reads: “The absolute cardiovascular predicted risk was similar between the laboratory-based and non-laboratory-based 2019 WHO cardiovascular risk models. **Pending validation from longitudinal studies, the non-laboratory-based model (instead of the laboratory-based which requires additional resources) could be used when assessing CVD risk in Peruvian population.**”

Q2. In the strength and limitations, do you consider the third point as a strength or limitation?

A2. This was considered as a limitation. For massive screening of hypertension, the WHO recommends taking 3 blood pressure measurements and using the mean of the last two measurements.¹ This was not possible with the data we had (because they only measured blood pressure twice, not three times).

For better understanding, these lines now read (p. 14): “Third, we only used one blood pressure record (the second SBP out of two measurements). Ideally, and following standard protocols

recommended by WHO¹ and other international organizations,² we should have used the average of multiple records having discarded the very first measurement. This was not possible with the available data because they only measured blood pressure twice. Nonetheless, we performed a sensitivity analysis using the mean SBP of the first and second records and had virtually the same results as in our main analysis.”

Q3. You stated that: “Our results suggest that the agreement between the laboratory-based and non-laboratory-based models was appropriate among Peruvians with low CVD risk and younger than 60 years” How did you show the appropriateness of this comparisons in your study?

A3. We used three statistical methods to determine the agreement between the laboratory- and non-laboratory-based models: Bland Altman plots (shown in Figures 1-5); Lin's concordance coefficient correlation (LCCC; shown in Table 2); and kappa statistics (shown in Supplementary Tables 3-9). In the total population, the agreement measured by all three methods was appropriate. For example, limits of agreement depicted in the Bland Altman plots ranged from -1.93 to 1.89 in men and from -1.23 to 1.39 in women. Nonetheless, when focusing on those with risk factors (e.g., those with diabetes), the agreement was less clear.

To address the reviewer's comment, we have reworded the referenced statement. New lines (p. 13) read: “According to the three statistical analyses we implemented (Bland-Altman plots, Lin's concordance coefficient correlations; and kappa statistics), our results suggest that the agreement between the laboratory-based and non-laboratory-based models was appropriate among Peruvians with low CVD risk and younger than 60 years”

Q4. Authors failed to show how justify the importance of this study, and the need for comparing the two algorithms in this context. Many studies have compared CVD algorithms. Why is the comparison of these two important, especially for your population?

A4. Many studies have compared different CVD algorithms and fewer studies have compared the two versions of the same algorithm (laboratory vs non-laboratory). Probably, because not all CVD algorithms have these two options (laboratory vs non-laboratory). We have included these lines in the introduction to further justify the importance of this study, especially for this population: “Because laboratory biomarkers (e.g., total cholesterol) may not be available in primary health care centres in LMICs limiting the use of laboratory-based CVD risk prediction models, the non-laboratory-based model arises as a handy tool for clinicians in LMICs. Similarly, CVD risk prediction models are used to monitor the prevalence of high CVD risk and treatment coverage (i.e., people at high CVD receiving treatment). In this context, countries conducting national or large population-based health surveys without lipid biomarkers, could benefit from the non-laboratory-based models. Peru, for example, does not have regular national health surveys including total cholesterol, but Peru has a yearly national health survey including anthropometrics, blood pressure, and health questionnaires. In Peru, and other similar LMICs, it would not be possible to monitor the burden of high CVD risk with a laboratory-based model and the non-laboratory-based model rises as the only alternative.”

Predicting absolute CVD risk is essential for CVD prevention, which are the major cause of mortality in LMICs. Although there are different CVD algorithms to compute absolute CVD risk, none has proved to produce reliable estimates for LMICs. The 2019 WHO CVD risk charts were adapted for LMICs using extensive datasets for its derivation, recalibration, and validation, which brings them several advantages over previous risk charts.

We have included the following lines in the Discussion (p. 14) to further justify the importance of the study: “We leveraged on a nationally-representative survey **conducted in a LMIC. Different**

¹ World Health Organization. STEPwise Approach to NCD Risk Factor Surveillance (STEPS) [Internet]. [cited 2022 Jan 10]. Available from: <https://www.who.int/teams/noncommunicable-diseases/surveillance/systems-tools/steps>

² Poulter NR, Borghi C, Damasceno A, Jafar TH, Khan N, Kokubo Y, et al. May Measurement Month 2019: results of blood pressure screening from 47 countries. *Eur Heart J Suppl.* 2021 May;23(Suppl B):B1–5.

CVD risk equations have been created but none has proved to produce reliable estimates for LMICs. The 2019 WHO CVD risk charts were adapted for LMICs using extensive datasets for its derivation, recalibration, and validation, which brings them several advantages over previous risk charts.”

Q5. There was little explanation for findings of this study in the discussion.

A5. We have added the following lines in the Discussion (pp. 14-15): “In our study, diabetes status was only included in the laboratory-based model (and not in the non-laboratory-based model) and reasonably, we observed the lowest agreement between both models in those with diabetes. That is, in people with diabetes, the non-laboratory-based model underestimated the absolute CVD risk computed following the laboratory-based model. Probably because people with diabetes are already at high CVD risk and the non-laboratory model we used without diabetes information would underestimate the absolute risk.”

REVIEWER #2

We appreciate **Dr Mirzaei** took the time to comment on our work. We have incorporated his suggestions.

Q1. Briefly explain how to sample the population in the national study.

A1. The CENAN's survey sample followed a probabilistic sampling design approach with two stages. First, clusters were randomly selected considering three strata: 1) Urban areas except Lima city, 2) Rural areas, and 3) Lima city. Then, households (of adults aged 18-59 years living in) were randomly selected within each cluster.

We have added the following lines in Methods addressing the reviewer's comment (p. 4): "The CENAN's survey sample followed a probabilistic sampling design approach with two stages. First, clusters were randomly selected considering three strata: 1) Urban areas except Lima city, 2) Rural areas, and 3) Lima city. Then, households (of adults aged 18-59 years living in) were randomly selected within each cluster."

Q2. The inclusion and exclusion criteria of the study should be stated. Not considering a history of previous cardiovascular disease confounds the final analysis. This group is at high risk. Although the limitations are mentioned, it is necessary to discuss more.

A2. We have added the selection criteria used for the survey sample in Methods. This now reads (p. 4): "To be selected for the survey sample, participants had to fulfil the following inclusion criteria: 1) Adults aged 18-59 years, and 2) Fasting 9-12 hours for blood biomarkers. The following participants were excluded: 1) Pregnant and postpartum women, 2) Adults taking medication that could alter glucose and lipid profiles, 3) Adults with congenital diseases that could limit anthropometrics measurement (e.g., Down syndrome)."

Regarding the limitation of not having excluded those with history of CVD, we have added the following lines (in bold) in the Discussion (p. 15): "Second, as CENAN's survey did not include history of cardiovascular events, we assumed all participants were free of CVD to use the 2019 WHO CVD risk score. This approach could have led to higher absolute cardiovascular risk because people who have had a cardiovascular event (e.g., myocardial infarction) are at higher risk of another cardiovascular event. **Nonetheless, considering that our study population was young and therefore with a low incidence of cardiovascular diseases,³ the proportion of people with history of CVD excluded from the total sample size would have been small; not excluding potentially a small group may not have altered the overall results.**"

Q3. How is the sample size calculated?

A3. The following formula was used in the CENAN's survey to compute its sample size. We have included this formula in Supplementary Figure 1:

$$n_h = \frac{N_h Z^2 P_h Q_h}{(N_h - 1)d^2 + Z^2 P_h Q_h} * TNR$$

Where:

N_h : number of people within an age group in the "h" conglomerate

n_h : number of people in the sample within an age group in the "h" conglomerate

d: error margin assumed in the estimation of P_h

Z: 95% confidence level

TNR (*Tasa de No Respuesta* in Spanish): Expected refusal rate

P_h : prevalence of overweight in adults in the "h" conglomerate

³ GBD Results Tool | GHDx [Internet]. Available from: <http://ghdx.healthdata.org/gbd-results-tool>

We have included the following lines in the Methods to address the reviewer's comment (p. 4): "The CENAN's survey sample was computed using the formula shown in Supplementary Figure 1 and followed a probabilistic sampling design approach with two stages..."

Q4. This study used national survey data. The approval of the ethics committee for the use of data should be noted.

A4. The CENAN's survey followed ethical guidelines. We have included the following lines in Methods (p. 4) addressing this: "**CENAN's survey adhered to ethical guidelines and followed a standardised protocol that has been published elsewhere.⁴ Each participant was informed about all procedures and techniques used in the survey; also, participants could have left the study at any time and their personal information was kept confidential.⁴**"

We opted not to seek approval of an Ethics Committee because this work was deemed as of minimal risk (we analyzed a dataset which is open access and available for reanalysis by any research group). We included the following lines (in bold) addressing the reviewer's comment (p. 7): "This study used de-identified nationally-representative survey data that can be requested from the CENAN and is used for independent analyses.⁵ **Authors had no access to the participants' personal information. Therefore, this work was deemed as of minimal risk** and we did not seek approval by an Ethics Committee to conduct the analysis."

Q5. In the discussion, it is necessary to compare the results of this study with similar studies.

A5. We have included a *Research in context* section (p. 14) in the Discussion, in which our results were compared with similar studies. The following lines address the reviewer's comment (p. 14): "The study most comparable to ours evaluated the agreement between the Framingham 10-year CVD risk laboratory and non-laboratory models on a population aged 40-75 years in southern Iran.⁶ They found the mean CVD risk following the non-laboratory-based model (9.4%) was higher than the laboratory-based model (6.7%). Additionally, their limits of agreement between both Framingham models in people <60 years old were wider compared to ours in both men (-1.9-1.9 by our estimates versus -2.5%-8.9% by Rezaei et al.) and women (-1.2-1.4 by our estimates versus -2.3%-4.6% by Rezaei et al.). This could be explained by the fact that Rezaei et al. included an older population, which tend to have higher levels of CVD risk factors and therefore higher absolute CVD risk. As limits of agreement between two models tend to widen with higher CVD risk, our limits of agreement would presumably be wider if we had studied a similar population to that of the work by Rezaei et al. The differences between our results could be further explained by the CVD risk score herein used. We used the 2019 WHO CVD risk models, whereas Rezaei et al. used the Framingham risk scores. The Framingham risk score was developed for a more specific population (Caucasians in the US), yet the 2019 WHO CVD risk model was developed and recalibrated for a global use (e.g., those living in LMICs)."

"The agreement between the 2019 WHO laboratory- and non-laboratory-based model was also explored in the global work convened by the WHO.⁷ They applied the two models to WHO STEPS surveys and compared the proportion of people categorized at different levels of predicted CVD risk.

⁴ Centro Nacional de Alimentación y Nutrición. ESTADO NUTRICIONAL EN ADULTOS DE 18 A 59 AÑOS, PERÚ: 2017 - 2018 [Internet]. 2021. Available from: https://web.ins.gob.pe/sites/default/files/Archivos/cenan/van/sala_nutricional/sala_3/2021/Informe%20Tecnico-%20Estado%20nutricional%20en%20adultos%20de%2018%20a%2059%20a%C3%B1os%20CVIANEV%202017-2018.pdf

⁵ Guzman-Vilca WC, Yovera-Juarez EA, Tarazona-Meza C, García-Larsen V, Carrillo-Larco RM. Sugar-Sweetened Beverage Consumption in Adults: Evidence from a National Health Survey in Peru. *Nutrients*. 2022 Jan 28;14(3):582.

⁶ Rezaei F, Seif M, Gandomkar A, Fattahi MR, Hasanzadeh J. Agreement between laboratory-based and non-laboratory-based Framingham risk score in Southern Iran. *Sci Rep*. 2021 May 24;11(1):10767.

⁷ WHO CVD Risk Chart Working Group. World Health Organization cardiovascular disease risk charts: revised models to estimate risk in 21 global regions. *Lancet Glob Health*. 2019 Oct;7(10):e1332–45.

Overall, they found moderate agreement between both models, and their discrepancy was attributed to poor performance of the non-laboratory-based model in people with diabetes. This finding is consistent with our results because we found the widest limits of agreement, the lowest LCCC, and the lowest categorical agreement in people with self-reported diabetes. When possible, it would seem reasonable to use the laboratory-based model in those who have diabetes.

Q6. Has the agreement between the two methods been reported in several studies in other regions? Has a study been published that reported the difference in the results of the two methods? If yes, has the study method been different? Comparison with the findings of others is recommended.

A6. We have compared our results with two previous reports in which: i) they compared laboratory-based and non-laboratory-based CVD risk prediction models; ii) used similar statistical analyses; and iii) made similar sub-groups comparisons (e.g., between with/without diabetes). To the best of our knowledge, there are not several studies in other regions looking at the same research questions with overall similar methods. Please, kindly refer to our previous answer. We have compared our findings with previous research in the *Research in context* section (p. 14) in the Discussion.

WHO CVD Risk Chart Working Group. World Health Organization cardiovascular disease risk charts: revised models to estimate risk in 21 global regions. *Lancet Glob Health*. 2019 Oct;7(10):e1332–45.